# Ultralong purely organic aqueous phosphorescence supramolecular polymer for targeted tumor cell imaging

Wei-Lei Zhou[1], Yong Chen[1], Qilin Yu 🔅 [1,2], Haoyang Zhang[1], Zhi-Xue Liu[1], Xian-Yin Dai[1], Jing-Jing Li[1] & Yu Liu[1✉]

Purely organic room-temperature phosphorescence has attracted attention for bioimaging but can be quenched in aqueous systems. Here we report a water-soluble ultralong organic room-temperature phosphorescent supramolecular polymer by combining cucurbit[n]uril (CB[7], CB[8]) and hyaluronic acid (HA) as a tumor-targeting ligand conjugated to a 4-(4-bromophenyl)pyridin-1-ium bromide (BrBP) phosphor. The result shows that CB[7] mediated pseudorotaxane polymer CB[7]/HA–BrBP changes from small spherical aggregates to a linear array, whereas complexation with CB[8] results in biaxial pseudorotaxane polymer CB[8]/HA–BrBP which transforms to relatively large aggregates. Owing to the more stable 1:2 inclusion complex between CB[8] and BrBP and the multiple hydrogen bonds, this supramolecular polymer has ultralong purely organic RTP lifetime in water up to 4.33 ms with a quantum yield of 7.58%. Benefiting from the targeting property of HA, this supramolecular polymer is successfully applied for cancer cell targeted phosphorescence imaging of mitochondria.

[1] College of Chemistry, State Key Laboratory of Elemento-Organic Chemistry, Nankai University, Tianjin 300071, P. R. China. [2] Key Laboratory of Molecular Microbiology and Technology, College of Life Sciences, Nankai University, Tianjin 300071, China. ✉email: yuliu@nankai.edu.cn

Room-temperature phosphorescence (RTP) emitted by purely organic molecules has been attracting increasing attention owing to its advantages over fluorescence, such as longer lifetime, larger Stokes shift, and the involvement of triplet states[1–6]. Such phosphorescent materials have therefore been widely used in organic light-emitting diodes[7,8], data-security[9,10], sensing[11,12], and bioimaging[13,14] applications, among others. RTP is usually achieved by means of crystalline packing[15,16] or by embedding phosphors in a rigid matrix[9,17,18]. For example, Tian et al[19]. developed amorphous metal-free phosphorescent materials by covalently attaching various phosphors to $\beta$-cyclodextrin ($\beta$-CD), the resulting materials exhibit efficient RTP emission arising from immobilization of the phosphors by a network of hydrogen bonds among the $\beta$-CDs. In addition, Kim et al[20]. reported a series of phosphor-containing metal-free organic materials that show enhanced RTP emission because molecular motion is restricted by covalent cross-linking between the phosphors and a polymer matrix. Recently, we reported a solid-state supramolecular phosphorescence material that is composed of cucurbit[6]uril (CB[6]) and 4-(4-bromophenyl)-1-methylpyridin-1-ium chloride[21] and that shows an excellent phosphorescence quantum yield (81.2%). In addition, its lifetime can be markedly enhanced (to 2.62 s) by replacing 4-(4-bromophenyl)-1-methyl-pyridin-1-ium chloride with 4-phenyl-1-methylpyridin-1-ium chloride[22].

Moreover, RTP could be expected to offer many advantages in vivo, given that it is readily distinguishable from spontaneous fluorescence and background fluorescence in cellular organelles[13,23]. Unfortunately, most systems showing RTP are solid-state, the practical utility of RTP in aqueous biosystems is limited due to the quenching of the oxygen and other molecules that occur in aqueous solution. Thus, the development of purely organic compounds that show RTP in aqueous solution is urgently needed. More recently, wu and co-workers[24] reported that difluoroboron-$\beta$-diketonate nanoparticles dispersed in water by hydrophobic agglomeration emit RTP (P$\tau$ = 29.0 µs) upon excitation with visible and near-infrared light. In addition, Tian and Zhu et al[25]. prepared an amphiphilic nano-assembly based on a monochromophoric polymer for the ratiometric tracing of hypoxia in vivo via oxygen-insensitive fluorescence emission and oxygen-dependent phosphorescence ($\tau$ = 7.96 µs) in aqueous solution. However, millisecond-level RTP from purely organic materials in water has rarely been reported.

Macrocyclic compounds (i.e., cyclodextrin, cucurbituril) have become a research hotspot for realizing purely organic phosphorescence in aqueous solution due to their special properties of internal hydrophobicity/external hydrophilicity and host–guest interactions[12,26–32]. Up to now, although the phosphorescence in water has made great progress through the host–guest interactions, the purely organic phosphorescence with long lifetime and function in water is rarely reported and still faces great opportunities and challenges. In this study, we constructed two supramolecular assemblies consisting of three components: cucurbit[n]urils (CB[n]s, where $n = 7$ or 8), which are biocompatible macrocycles that strongly bind organic cations[33–35]; hyaluronic acid (HA), a water-soluble, biocompatible, biodegradable polymer that is specifically recognized by receptors (e.g., CD44 and RHAMM) overexpressed on the surface of cancer cells[36,37]; and 4-(4-bromophenyl)-pyridin-1-ium (BrBP), an organic phosphor (Fig. 1). Intriguingly, the biaxial pseudorotaxane polymer CB[8]/HA–BrBP exhibited RTP with an ultralong lifetime (4.33 ms) and a high quantum yield (7.58%) in aqueous solution. These results were attributed to strong binding between CB[8] and BrBP, as well as the hydrogen-bond networks of the HA polymers, which promoted intersystem crossing (ISC), restricted the molecular motion and minimized collision of the phosphor triplet state with triplet oxygen and other molecules. We found that this RTP pseudorotaxane polymer was capable of targeting cancer cells, especially imaging in the mitochondrion. The work reported herein not only opens an important avenue for the development of purely organic materials that exhibit RTP in aqueous solution but also extends the applications of RTP in such solution.

## Results

**Binding of CBs and BrBP–NH2 and optical properties of the resulting complexes.** To explore the effect of host–guest complexation between CBs and phosphors on RTP emission in aqueous solution, we used $^1$H NMR spectroscopy, UV–vis spectroscopy, and isothermal titration calorimetry (ITC) to elucidate the binding behaviors of CB[7] and CB[8] with a model phosphor, 1-(3-aminopropyl)-4-(4-bromophenyl)pyridine-1-ium bromide hydrobromide (BrBP–NH$_2$). The synthesis of BrBP–NH$_2$ was shown in the Supporting Information (Supplementary Fig. 1). Upon addition of CB[7] to BrBP–NH$_2$, the $^1$H NMR signals of the aromatic protons of BrBP–NH$_2$ at 6.5–9.0 ppm (H$_{a–c}$) exhibited marked upfield shifts, whereas the signals of the alkyl chain protons at 2.3–4.8 ppm (H$_{d–f}$) remained almost completely unchanged (Supplementary Fig. 2). UV–vis spectroscopy showed that the absorption band of BrBP–NH$_2$ at 308 nm gradually decreased in intensity and underwent a slight bathochromic shift as increasing amounts of CB[7] were added (Fig. 2a). Moreover, two apparent isosbestic points (at 250 and 325 nm) were also observed. These changes indicate that the BrBP moiety became encapsulated in the CB[7] cavity. In addition, the stoichiometry and the association constants ($K_s$) for binding between BrBP–NH$_2$ and CBs were determined by ITC and optical analyses. In the ITC titrations (Supplementary Fig. 3A), the calculations were repeated as a 1:1 complex formation, and the titration data could be fitted well with a model characterized by one set of binding sites, giving a $K_s$ value of $(3.81 \pm 0.22) \times 10^6$ M$^{-1}$. Taken together, our results indicate that BrBP–NH$_2$ and CB[7] formed a pseudo[2]rotaxane.

In the case of the CB[8]/BrBP–NH$_2$ system, the $^1$H NMR signals for the protons of the alkyl chain of free BrBP–NH$_2$ at 2.3–3.3 ppm (H$_{e,f}$) were shifted slightly downfield upon addition of CB[8], but the signals of the aromatic protons (H$_{a–c}$) shifted upfield (Supplementary Fig. 2). As the amount of CB[8] was increased, the UV–vis adsorption band of BrBP–NH$_2$ at 300 nm gradually decreased in intensity and became slightly red-shifted, and these changes were accompanied by the appearance of two apparent isosbestic points (at 250 and 325 nm, Fig. 2e). The ITC titration data could be fitted well with a model characterized by two successive binding sites (Supplementary Fig. 3B), and the $K_{a1}$ and $K_{a2}$ values for the two sites were calculated to be $(4.49 \pm 0.19) \times 10^5$ M$^{-1}$ and $(2.43 \pm 0.08) \times 10^6$ M$^{-1}$, respectively. These results point to the formation of a more stable 1:2 inclusion complex between CB[8] and BrBP–NH$_2$ with a high total association constant, up to $(1.09 \pm 0.01) \times 10^{12}$ M$^{-2}$, in which strong π–π stacking interactions between the two BrBP–NH$_2$ moieties resulting in the flexible alkyl chain near the entrance of the CB[8] cavity.

Interestingly, the photoluminescence spectra of both CB[7]/BrBP–NH$_2$ and CB[8]/BrBP–NH$_2$ showed fluorescence at about 380 nm (Fig. 2b, c, f, and g) as well as phosphorescence at about 500 nm (Fig. 2b, d, f, and h); and the latter was verified by means of an oxygen quenching experiment. In a control experiment, no appreciable phosphorescence emission was observed for free BrBP–NH$_2$ even under N$_2$ (Fig. 2f). Notably, the phosphorescence of CB[8]/BrBP–NH$_2$ was 33 times as strong as that of CB[7]/BrBP–NH$_2$ under the same conditions (Supplementary Fig. 4).

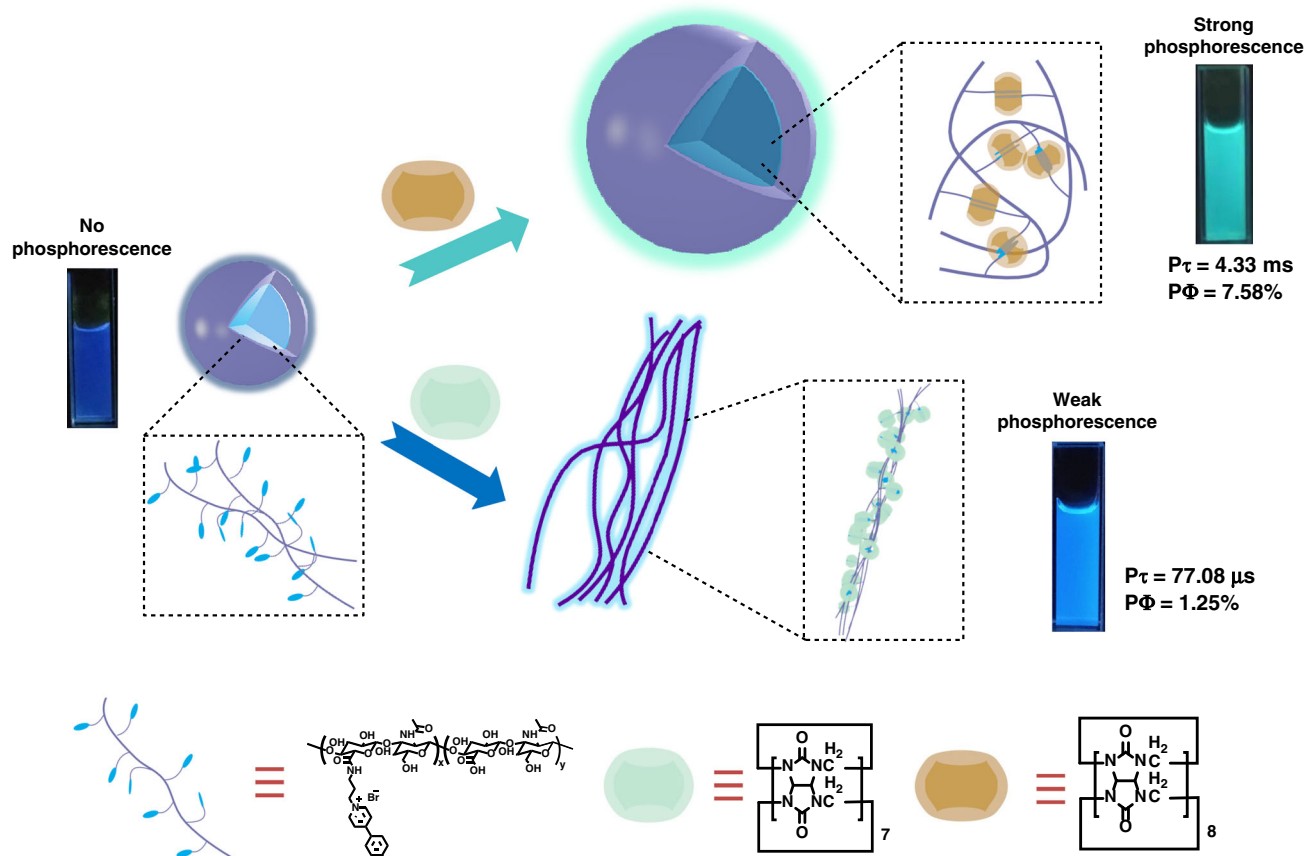

**Fig. 1 Schematic illustration.** The construction and behavior of CBs/HA–BrBP supramolecular pseudorotaxane polymers in aqueous solution.

Furthermore, time-resolved fluorescence decay curves were measured (Supplementary Fig. 5, Table 1), and the lifetimes of the emissions at 380 nm for BrBP–NH$_2$, CB[7]/BrBP–NH$_2$, and CB[8]/BrBP–NH$_2$ were determined to be 217.21, 300.13, and 226.56 ps. In contrast, the emission at 500 nm for CB[7]/BrBP–NH$_2$ had a lifetime on the order of microseconds (55.31 µs) and a quantum yield of 1.07%; and unexpectedly, the lifetime of the 500 nm emission for CB[8]/BrBP–NH$_2$ was 1.54 ms with a quantum yield of 2.79%.

**Optical properties of CBs in complex with polymer HA–BrBP.** Inspired by these results, we speculated that polypseudorotaxanes based on BrBP-modified HA and CBs might also emit relatively strong phosphorescence due to the hydrogen-bonding interactions among the HA polymers and to host–guest interactions between the CBs and BrBP. Therefore, we synthesized HA–BrBP by a two-step procedure involving an amide condensation reaction between HA (250 kDa) and BrBP–NH$_2$. Grafting of BrBP groups onto the HA chain was confirmed by ¹H NMR spectroscopy (Supplementary Fig. 6): the signals at 7.5–9.0 ppm were assigned to the aromatic protons of BrBP, and the signals at 2.0 and 2.7–4.5 ppm were assigned to protons of HA and the alkyl chain of BrBP. On the basis of the ratio of the integrations of the methylene protons adjacent to nitrogen atom of the pyridine ring and the *N*-acetyl protons of HA at 2.0 ppm, the degree of substitution by BrBP groups in the HA–BrBP was calculated to be 3.5%, indicating that one of every 28.6 polysaccharide units was modified by a BrBP group. Therefore, HA–BrBP obtained in this way was deemed to be suitable for specific binding to the CD44 and RHAMM receptors that are overexpressed on cancer cells.

Upon addition of CB[7] or CB[8] to HA–BrBP in water, the band at 250–350 nm in the UV–vis spectrum of free HA–BrBP gradually decreased in intensity and exhibited a slight red shift (Fig. 3a). In addition, two isosbestic points (at 275 and 325 nm) were observed. This behavior was similar to that observed for the CBs/BrBP–NH$_2$ complexes, indicating that CBs and HA–BrBP readily formed pseudorotaxane polymers. Upon addition of CB[8] to HA–BrBP, emission peaks at around 380 and 500 nm were observed, and the intensity of the emission at 500 nm was about two times as that of the emission at around 380 nm (Fig. 3b–d). In contrast, the CB[7]/HA–BrBP complex showed an emission peak at around 380 nm but only a relatively weak shoulder at around 500 nm (Fig. 3b). Notably, the intensity of the CBs/HA–BrBP emission at around 500 nm was enhanced when N$_2$ was bubbled into the solution, indicating that this emission should be assigned to phosphorescence (Fig. 3b–d). The time-resolved (delayed by 0.2 ms) photoluminescence spectra also exhibited strong phosphorescence emission by CB[8]/HA–BrBP at 500 nm (Fig. 3d), and the phosphorescence of CB[8]/HA–BrBP was much stronger than that of CB[7]/HA–BrBP (Fig. 3c) or CB[8]/BrBP–NH$_2$ (Fig. 2h). Analysis of time-resolved decay data (Fig. 4, Supplementary Fig. 7, Table 1) showed that the fluorescence lifetimes of HA–BrBP, CB[7]/HA–BrBP, and CB[8]/HA–BrBP at 380 nm were on the order of nanoseconds. However, the phosphorescence lifetime of CB[7]/HA–BrBP at 500 nm was 77.08 µs, and that of CB[8]/HA–BrBP at 500 nm was ultralong for 4.33 ms (quantum yield 7.58%); and under N$_2$, the lifetime of the latter increased to 5.03 ms (quantum yield 8.77%). Taken together, these results demonstrated that highly efficient RTP in aqueous solution could be achieved with the CB[8]/ HA–BrBP pseudorotaxane polymer. To the best of our

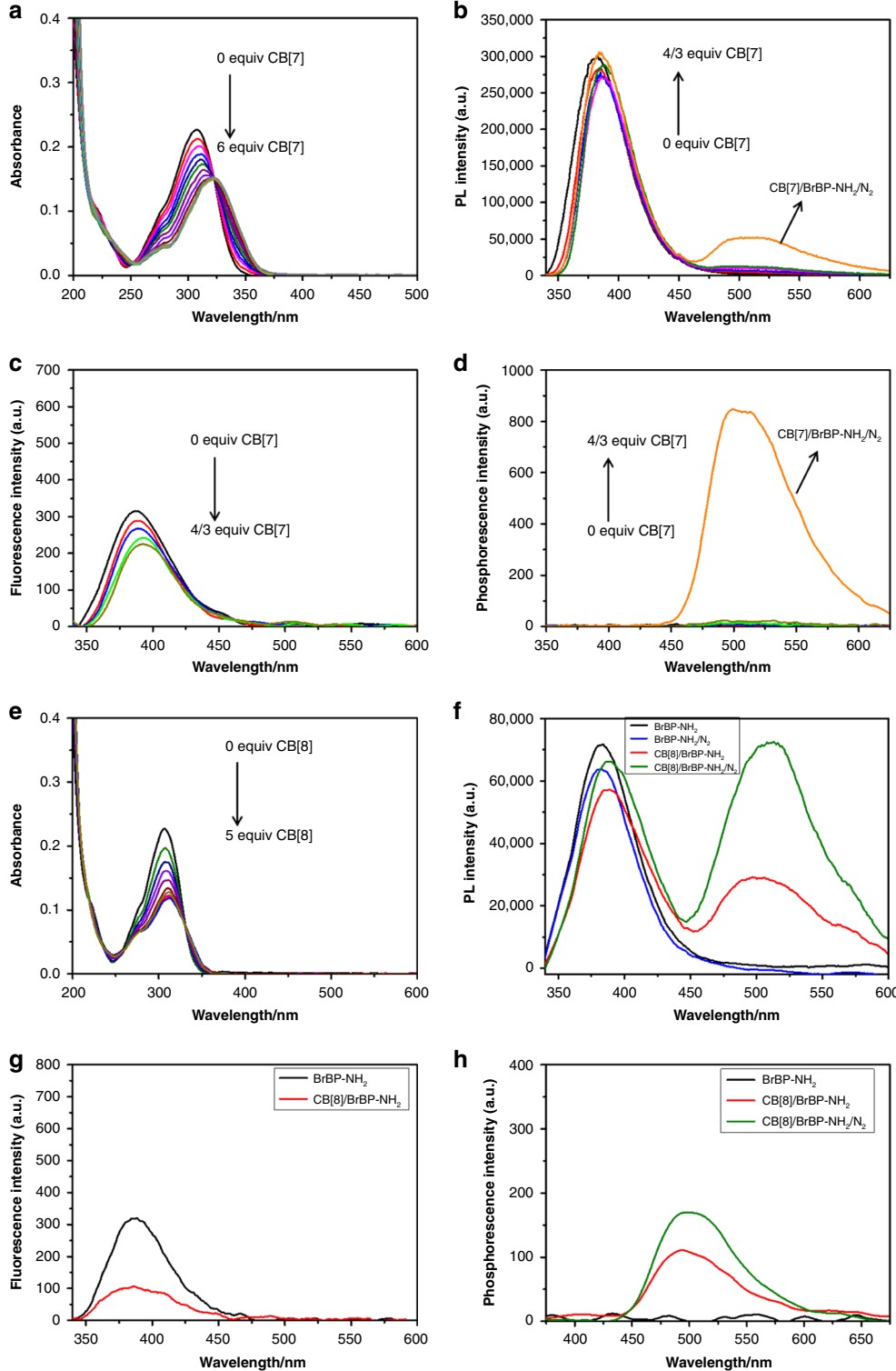

**Fig. 2 Effect of complexation with CB[7] and CB[8] on the spectra of BrBP–NH₂.** **a** Absorption spectra of BrBP–NH₂ (0.01 mM) in the absence (black) and presence (yellow) of CB[7] (0.06 mM) in water at 25 °C. **b** Prompt photoluminescence spectra, **c** fluorescence spectra, and **d** phosphorescence spectra (delayed by 0.2 ms, Ex. Slit = 10 nm, Em. Slit = 10 nm) of BrBP–NH₂ (black), CB[7]/BrBP–NH₂ (yellow), and CB[7]/BrBP–NH₂/N₂ (orange) ([BrBP–NH₂] = 0.5 mM, [CB[7]] = 0.67 mM) in water at 25 °C (λ$_{ex}$ = 320 nm). **e** Absorption spectra of BrBP–NH₂ (0.01 mM) in the absence (black) and presence (red) of CB[8] (0.05 mM) in water at 25 °C. **f** Prompt photoluminescence spectra, **g** fluorescence spectra, and (**h**) phosphorescence spectra (delayed by 0.2 ms, Ex. Slit = 5 nm, Em. Slit = 5 nm) of BrBP–NH₂ (black), CB[8]/BrBP–NH₂ (red), and CB[8]/BrBP–NH₂/N₂ (green) ([BrBP–NH₂] = 0.5 mM, [CB[8]] = 0.25 mM) in water at 25 °C (λ$_{ex}$ = 320 nm).

**Table 1 Photophysical data for pseudorotaxanes and their constituents.**

| compound | $\tau$ (380 nm) | $\tau$ (500 nm) | $\Phi$ (500 nm) | $k_{isc}{}^a$ ($s^{-1}$) | $k_r^{Phos\,b}$ ($s^{-1}$) | $k_{nr}^{Phos\,c}$ ($s^{-1}$) |
|---|---|---|---|---|---|---|
| BrBP–NH$_2$ | 217.21 ps | 0 | 0 | – | – | – |
| CB[7]/BrBP–NH$_2$ | 300.13 ps | 55.31 µs | 1.07% | $3.57 \times 10^7$ | $1.93 \times 10^2$ | $1.79 \times 10^4$ |
| CB[8]/BrBP–NH$_2$ | 226.56 ps | 1.54 ms | 2.79% | $1.23 \times 10^8$ | $1.81 \times 10$ | $6.31 \times 10^2$ |
| HA-BrBP | 200.89 ps | 0 | 0 | – | – | – |
| CB[7]/HA–BrBP | 437.27 ps | 77.08 µs | 1.25% | $2.86 \times 10^7$ | $1.62 \times 10^2$ | $1.28 \times 10^4$ |
| CB[7]/HA–BrBP/N$_2$ | 424.73 ps | 124.57 µs | 1.49% | – | – | – |
| CB[8]/HA–BrBP | 201.47 ps | 4.33 ms | 7.58% | $3.76 \times 10^8$ | $1.75 \times 10$ | $2.13 \times 10^2$ |
| CB[8]/HA–BrBP/N$_2$ | 213.29 ps | 5.03 ms | 8.77% | – | – | – |

The concentrations of BrBP–NH$_2$, CB[7] and CB[8] were 0.5, 0.5 and 0.25 mM, respectively.
[a]The intersystem crossing rate constant $k_{isc} = \Phi_{Phos}/\tau_{Fluo}$.
[b]The radiative decay rate constant of phosphorescence $k_r^{Phos} = \Phi_{Phos}/\tau_{Phos}$.
[c]The nonradiative decay rate constant of phosphorescence $k_{nr}^{Phos} = (1-\Phi_{Phos})/\tau_{Phos}$.

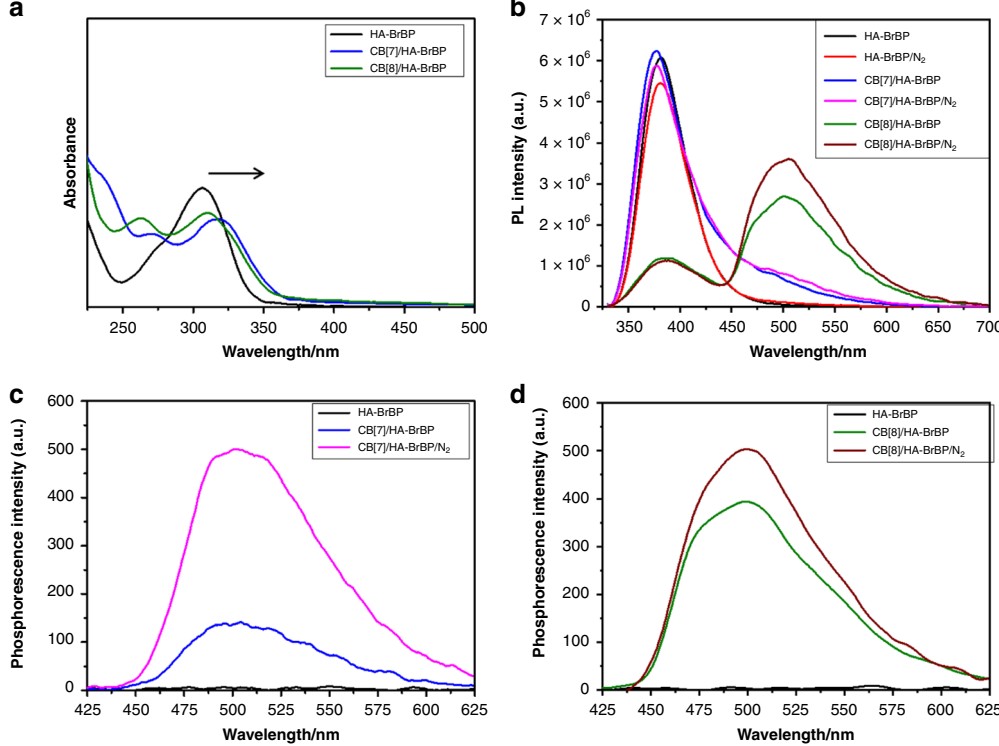

**Fig. 3 Effect of complexation with CB[7] and CB[8] on the spectra of HA–BrBP. a** UV–vis absorption spectra of HA–BrBP, CB[7]/HA–BrBP, and CB[8]/ HA–BrBP in aqueous solution at 25 °C ([BrBP] = 0.05 mM, [CB[7]] = 0.05 mM, [CB[8]] = 0.025 mM). **b** The Prompt photoluminescence spectra of HA–BrBP (0.5 mM) (black), HA–BrBP/N$_2$ (red), CB[7]/HA–BrBP (light blue), CB[7]/HA–BrBP/N$_2$ (pink), CB[8]/HA–BrBP (green), and CB[8]/HA–BrBP/ N$_2$ (wine) in aqueous solution at 25 °C. **c** The phosphorescence spectra of HA–BrBP, CB[7]/HA–BrBP, and CB[7]/HA–BrBP/N$_2$ at 298 K in aqueous solution (delayed by 0.2 ms, Ex. Slit = 10 nm, Em. Slit = 10 nm). **d** The phosphorescence spectra of HA–BrBP, CB[8]/HA–BrBP, and CB[8]/HA–BrBP/N$_2$ at 298 K in aqueous solution (delayed by 0.2 ms, Ex. Slit = 5 nm, Em. Slit = 5 nm). ([BrBP] = 0.5 mM, [CB[7]] = 0.5 mM, [CB[8]] = 0.25 mM).

knowledge, this polymer exhibits the relatively longer-lived RTP of any purely organic material in aqueous solution reported to date (Supplementary Table 1). Moreover, we further tested the CB[6]/HA–BrBP system and found that the phosphorescence of CB[6]/HA–BrBP was a little stronger than that of CB[7]/ HA–BrBP but much weaker than that of CB[8]/HA–BrBP under the same condition (Supplementary Fig. 8). Considering the fairly lower water solubility of CB[6] than CB[7] and CB[8] under our experimental condition[33,34], we chose CB[7] and CB[8] to conduct systematic optical research.

Subsequently, we investigated the effect of temperature on the phosphorescence of CBs/HA–BrBP in aqueous solution (Supplementary Fig. 9). As the temperature was decreased from 298 to

100 K, the photoluminescence intensities of CB[7]/HA–BrBP and CB[8]/HA–BrBP at 500 nm increased 10-fold and fourfold, and their phosphorescence lifetimes increased markedly, to 17.19 and 16.94 ms, respectively (Supplementary Fig. 10), because nonradiative relaxation of triplets to the ground state was suppressed at low temperature[8]. In a control experiment, no HA–BrBP phosphorescence was observed in the absence of CBs in aqueous solution, even under N$_2$ (Fig. 3b black line/red line, Supplementary Fig. 11a, Fig. 4d black line). However, the photoluminescence of HA–BrBP at 500 nm gradually increased, while the corresponding lifetime increased from 0 to 18.46 ms, with the temperature decreasing from 298 to 100 K (Supplementary Fig. 11, Fig. 4d), because the vibrational loss was effectively

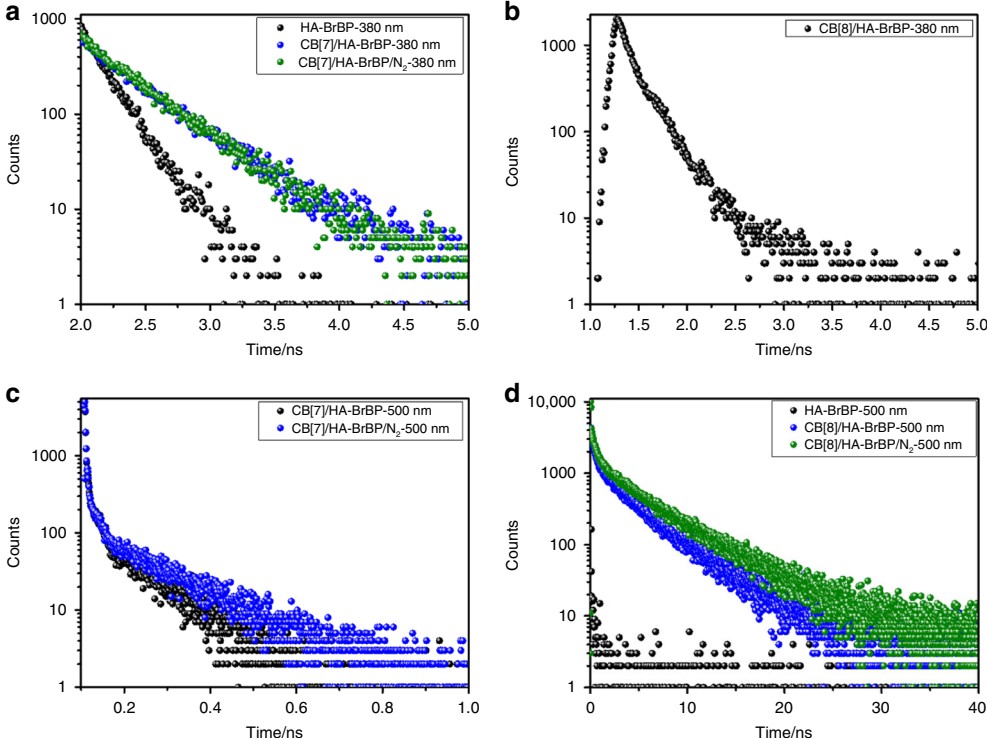

**Fig. 4 Fluorescence and phosphorescence lifetime contrast curves for CB[7]/HA–BrBP and CB[8]/HA–BrBP.** The fluorescence decay curves of (**a**) HA–BrBP, CB[7]/HA–BrBP, CB[7]/HA–BrBP/N$_2$ and (**b**) CB[8]/HA–BrBP at 380 nm at 298 K; The phosphorescence decay curves of (**c**) CB[7]/HA–BrBP, CB[7]/HA–BrBP/N$_2$ and (**d**) HA–BrBP, CB[8]/HA–BrBP, CB[8]/HA–BrBP/N$_2$ at 500 nm at 298 K. ([BrBP] = 0.5 mM, [CB[7]] = 0.5 mM, [CB[8]] = 0.25 mM).

suppressed at low temperature. Moreover, the photoluminescence intensities and the lifetime of CBs/HA–BrBP (peak at 500 nm) were higher than those of CBs/BrBP–NH$_2$. Taken together, these results indicate that both CBs and HA played important roles in the long-lived RTP exhibited by CB[8]/HA–BrBP.

**Mechanism of CBs/HA–BrBP phosphorescence**. To understand the unique phosphorescence of CB[8]/HA–BrBP, we performed density functional theory calculations to CB[8]/BrBP–NH$_2$ with the Gaussian 16 program (see "Methods" for details). Based on structural details of the geometry optimization results (Fig. 5a)[15,18,38], analysis of the frontier molecular orbitals (Supplementary Fig. 12) indicated that the lower energy gap between the orbitals of CB[8]/BrBP–NH$_2$ relative to that of BrBP–NH$_2$ was due to π–π stacking of BrBP–NH$_2$ in the CB[8]' cavity. According to the DFT geometry optimization results (Fig. 5a) and the noncovalent interaction (NCI) analysis (Supplementary Fig. 13), the Br atom of one BrBP–NH$_2$ moiety was suitably positioned relative to the pyridine ring of the other BrBP–NH$_2$ moiety (3.39 Å) to permit Br–π bonding[10]. In addition, there are obvious hydrogen bonds between the carbonyl of CB[8] and the hydrogen of pyridine ring (2.24 Å) as well as the NH$_2$ of BrBP–NH$_2$ (2.17 Å). Noteworthy, there are also halogen bonds between the C–Br of BrBP and the N atom of the adjacent BrBP (C–Br⋯N angle of 159.6°, Br⋯N distance 3.09 Å). The Mulliken charge (Supplementary Table 2) on the Br atom increased after the formation of aggregate, indicated that the C–Br⋯N halogen bonding changed the charge distribution on Br atom, which would affect the heavy atom effect. Furthermore, the supramolecular polymer CB[8]/HA–BrBP showed the decent phosphorescence quantum yield (7.58%) indicated that the encapsulation of CB[8], π–π/Br–π interaction, halogen bonding and the multiple hydrogen bonding jointly contribute to the long RTP of CB[8]/HA–BrBP in aqueous solution. The radiative and nonradiative decay rate

constants could be calculated according to the standard methods using the measured quantum yields and lifetimes of these assemblies, as shown in Table 1[22,39–41]. Indeed, the intersystem crossing rate ($k_{isc}$) of CB[8]/HA–BrBP was $3.76 \times 10^8\,\mathrm{s^{-1}}$, which was higher than that of CB[8]/BrBP–NH$_2$ ($1.23 \times 10^8\,\mathrm{s^{-1}}$) or CB[7]/HA–BrBP ($2.86 \times 10^7\,\mathrm{s^{-1}}$). And the radiative decay rate of phosphorescence ($k_r^{Phos}$) of CB[8]/HA–BrBP was $1.75 \times 10\,\mathrm{s^{-1}}$, which was lower than that of CB[8]/BrBP–NH$_2$ ($1.81 \times 10\,\mathrm{s^{-1}}$) or CB[7]/HA–BrBP ($1.62 \times 10^2\,\mathrm{s^{-1}}$). The nonradiative decay rate of phosphorescence ($k_{nr}^{Phos}$) complied with the same regular. On the basis of the Eqn. (Fig. 5), $\tau$ is inversely proportional to ($k_r^{Phos} + k_{nr}^{Phos}$), whereas $\Phi$ depends on efficient ISC (high $\Phi_{isc}$) and high efficiency of phosphorescence [$k_r^{Phos}/(k_r^{Phos} + k_{nr}^{Phos})$]. Accordingly, the ($k_r^{Phos} + k_{nr}^{Phos}$) of CB[8]/HA–BrBP, CB[8]/BrBP–NH$_2$, CB[7]/HA–BrBP and CB[7]/BrBP–NH$_2$ were $2.31 \times 10^2\,\mathrm{s^{-1}}$, $6.49 \times 10^2\,\mathrm{s^{-1}}$, $1.30 \times 10^4\,\mathrm{s^{-1}}$ and $1.81 \times 10^4\,\mathrm{s^{-1}}$ respectively, which indicated that the ($k_r^{Phos} + k_{nr}^{Phos}$) of CB[8]/HA–BrBP and CB[8]/BrBP–NH$_2$ were respectively smaller than that of CB[7]/HA–BrBP and CB[7]/BrBP–NH$_2$, as well as the ($k_r^{Phos} + k_{nr}^{Phos}$) of CB[8]/HA–BrBP and CB[7]/HA–BrBP were respectively smaller than that of CB[8]/BrBP–NH$_2$ and CB[7]/BrBP–NH$_2$. Meanwhile, the [$k_r^{Phos}/(k_r^{Phos} + k_{nr}^{Phos})$] of CB[8]/HA–BrBP, CB[8]/BrBP–NH$_2$, CB[7]/HA–BrBP and CB[7]/BrBP–NH$_2$ were 0.076, 0.028, 0.013 and 0.011 respectively. The [$k_r^{Phos}/(k_r^{Phos} + k_{nr}^{Phos})$] of CB[8]/HA–BrBP is the largest value of the four assemblies, which reveals the same regular. This further demonstrates that CBs and HA played a key role for achieving the long-lived phosphorescence in this process. A possible mechanism for the phosphorescence of CBs/HA–BrBP is illustrated in Fig. 5b. This mechanism has four important features: first, in aqueous solution, CBs provide a hydrophobic environment that protects the triplet state from the collision of the triplet oxygen and other molecules. Second, the strong host–guest interactions

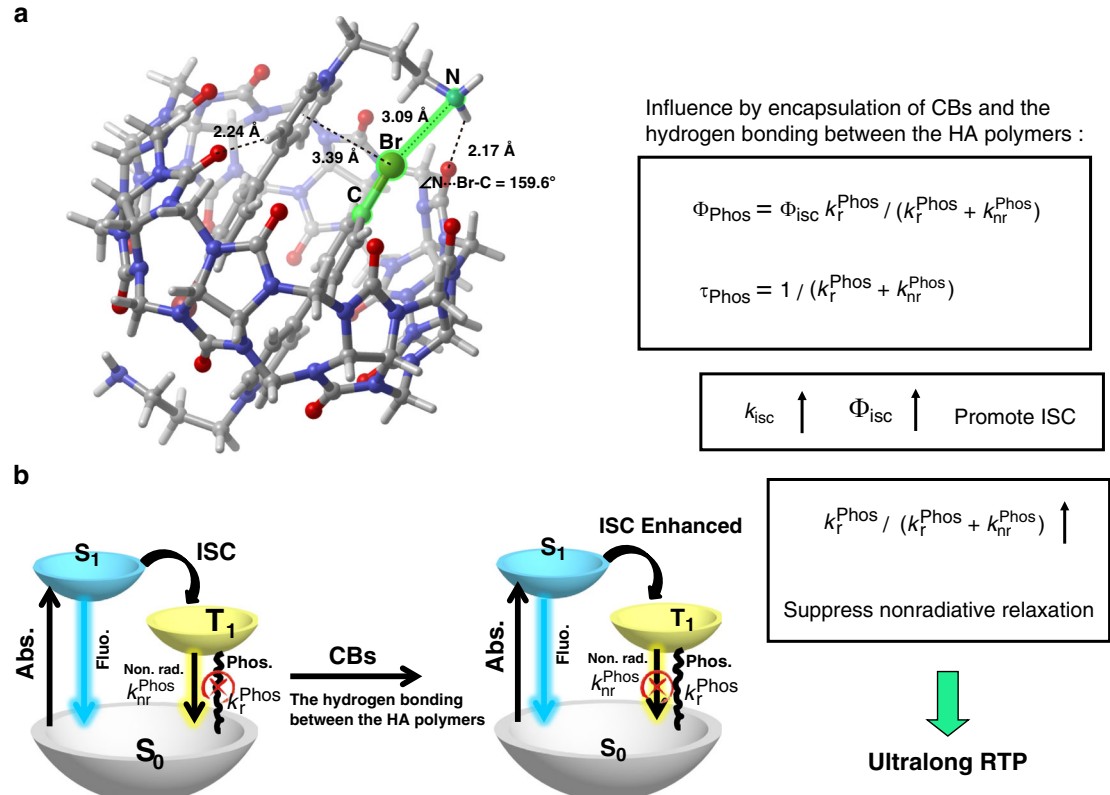

**Fig. 5 Mechanism of CBs/HA–BrBP phosphorescence. a** Structure of CB[8]/BrBP–NH$_2$ assembly as optimized by density functional theory calculations and **b** proposed mechanism of the long-lived RTP exhibited by this assembly.

between CB and BrBP(s), as well as the multiple hydrogen bonding, lock the BrBP unit(s) in different directions, probably restricting the molecular motion and reducing the non-radiative decay. Third, comparing with CB[7], CB[8] can accommodate two BrBP moieties, which results in more stable aggregation of CB[8]-enhanced π–π complexes (with a higher association constant) to limit molecular rotation and enhance ISC more efficiently, resulting in a longer phosphorescence lifetime. Fourth, the formation of halogen bonding between the C-Br of BrBP and the N atom of the adjacent BrBP (C-Br···N angle of 159.6°, Br···N distance 3.09 Å) increase the Mulliken charge on the Br atom and thus affected the heavy effect. As a result, the combination of the host–guest interaction, π–π/Br–π interaction, halogen bonding, and multiple hydrogen bonding jointly contribute to the long RTP of CB[8]/HA–BrBP in aqueous solution probably by restricting the molecular motion, promoting the ISC and reducing the non-radiative decay[14,17,31–34,37,42–45].

**Topological morphology of CBs/HA–BrBP.** We used transmission electron microscopy and scanning electron microscopy to obtain morphological information (Supplementary Fig. 14). HA–BrBP, CB[7]/HA–BrBP, and CB[8]/HA–BrBP existed as small spherical aggregates (average diameter, ~90 nm), nanofibers, and relatively large spherical aggregates (average diameter, ~200 nm), respectively. In addition, we measured the zeta potentials of HA–BrBP, CB[7]/HA–BrBP, and CB[8]/HA–BrBP to be −72.1, −52.4, and −31.7 mV, respectively (Supplementary Fig 15). The negatively charged surfaces were due to the carboxyl groups of the HA molecules and can be expected to give the assemblies a long circulation time and to reduce nonspecific cellular uptake, which should facilitate targeting[46].

**Phosphorescence imaging with the pseudorotaxane polymer CB [8]/HA–BrBP.** We investigated the utility of CB[8]/HA–BrBP for

phosphorescence imaging in living cells. The cells were incubated with CB[8]/HA–BrBP, and then the intracellular emission optical signal at 515–540 nm, which just maches the phosphorescence emission of the pseudorotaxane polymer, was acquired by confocal microscopy. As shown in Fig. 6a, all three types of cancer cells (A549, HeLa, KYSE-150) emitted strong green phosphorescence, whereas no obvious phosphorescence was observed in the human embryonic kidney cells (293T). These results demonstrated that the pseudorotaxane polymer preferentially targeted tumor cells over normal cells. In addition, the colocalization analysis suggested that the bright green phosphorescence of CB[8]/HA–BrBP in A549 cells displayed entire-overlapping with the mitochondrion marker MitoTracker Red as shown with the appearance of the merged yellow dyeing site, while the A549 cells incubated with HA–BrBP showed the fairly weak phosphorescence (Fig. 6b, c). In addition, a standard CCK-8 assay was used to evaluate the cytotoxicity of the pseudorotaxane polymer (Supplementary Fig. 16). The assay results showed that the viability of A549 and 293T cells was not significantly affected after incubation with CB[8]/HA–BrBP (0–100 μM) for 12 h, implying that CB[8]/HA–BrBP had low cytotoxicity. Thus, this imaging method involving pseudorotaxane polymer showed promising utility for phosphorescence imaging of mitochondrion in tumor cells. Furthermore, we also investigated the effect of the different HA polymer distributions on the detection performance of aggregates by changing either the molecular weight of the HA polymer skeleton or the ratios between CB[8] and BrBP moiety. The results show that, for the aggregates obtained from HA with molecular weights of 3, 100, and 1000 kDa (named CB[8]/HA$_{3k}$–BrBP, CB[8]/HA$_{100k}$–BrBP and CB[8]/HA$_{1000k}$–BrBP respectively), the detection abilities of aggregates towards A549 cells enhanced with the increase of the molecular weight of the HA polymer skeleton, i.e. CB[8]/HA$_{3k}$–BrBP < CB[8]/HA$_{100k}$–BrBP < CB[8]/HA$_{1000k}$–BrBP.

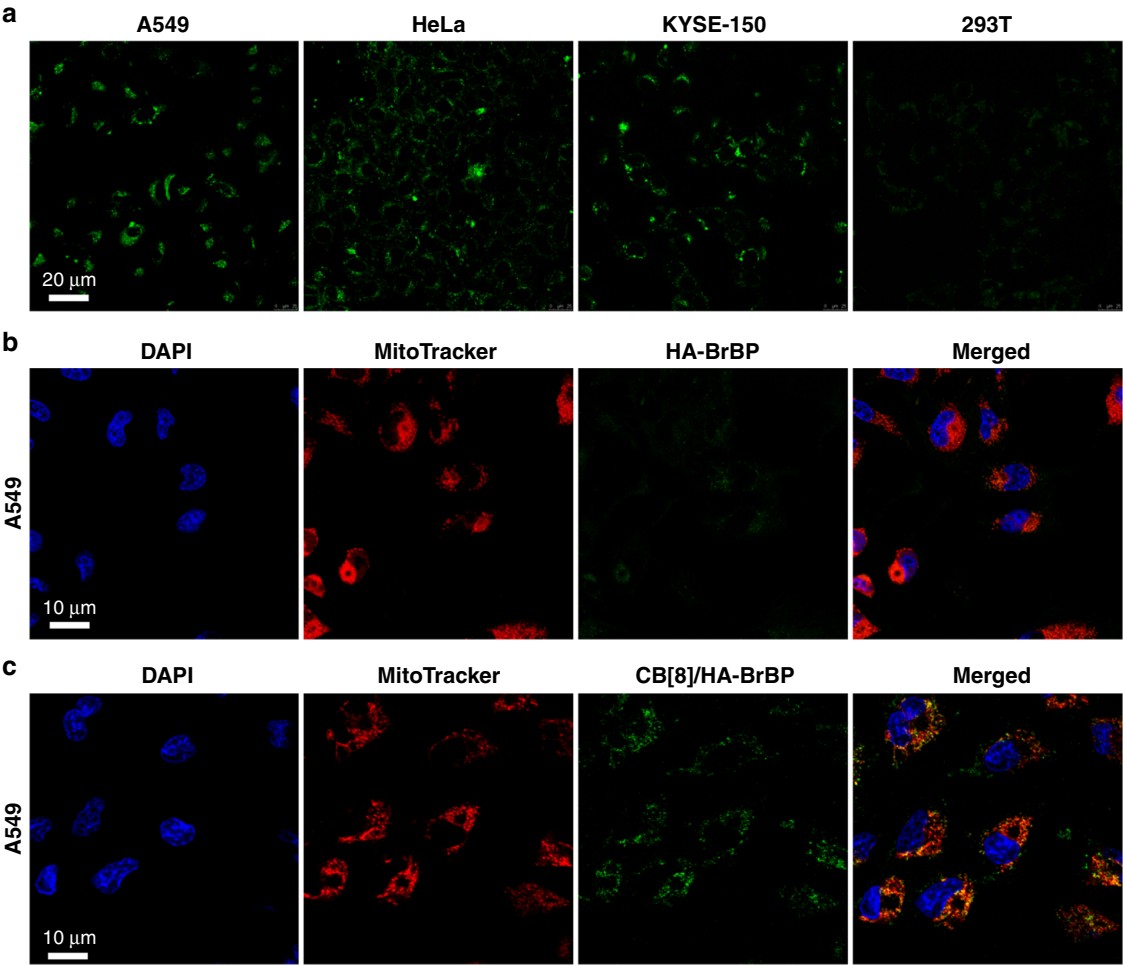

**Fig. 6 Confocal microscopy images. a** A549, HeLa, KYSE–150 and 293T cells incubated with CB[8]/HA–BrBP ([BrBP] = 25 μM, [CB[8]] = 12.5 μM).
**b** A549 cells incubated with HA–BrBP ([BrBP] = 25 μM). **c** A549 cells incubated with CB[8]/HA–BrBP ([BrBP] = 25 μM, [CB[8]] = 12.5 μM). 4,
6-Diamidino–2-phenylindole (DAPI, blue) was used to stain the nuclei, and MitoTracker (red) was used to stain the mitochondria.

However, all of CB[8]/HA$_{3k}$–BrBP, CB[8]/HA$_{100k}$–BrBP and CB[8]/HA$_{1000k}$–BrBP displayed the very weak detection abilities towards normal human cells (Supplementary Fig. 17, 18). For the aggregates obtained from the same HA polymer skeleton and the various CB[8]: BrBP ratios (CB: BrBP = 0 equiv. CB[8]: 1 equiv. BrBP, 0.09 equiv. CB[8]: 1 equiv. BrBP and 0.5 equiv. CB[8]: 1 equiv. BrBP), the phosphorescence intensity and cell imaging effect gradually increased with the increase of the CB: BrBP ratio, i.e., free HA–BrBP (CB[8]:BrBP = 0 equiv. CB[8]: 1 equiv. BrBP) < CB[8]/HA–BrBP at 0.09 equiv. CB[8]: 1 equiv. BrBP < CB[8]/HA–BrBP at 0.5 equiv. CB[8]: 1 equiv. BrBP (Supplementary Fig. 19). Considering that the use of UV-light as an excitation light might be harmful to biosystem, we investigated the possibility of achieving the phosphorescence emission and the phosphorescence imaging under the excitation of near-infrared light or visible light via the the up-conversion luminescence method (Supplementary Fig. 20). As shown in Supplementary Fig. 20b, after the addition of the up-conversion nanoparticles (UCNPs) to the supramolecular polymer, the photoluminescence spectrum of supramolecular polymer in water showed a clear emission peak at 510 nm assigned to the phosphorescence emission of CB[8]/HA–BrBP when excited by near-infrared light (980 nm). In the control experiment, the UCNPs showed no any appreciable emission over 500 nm under the same condition. Moreover, we also carried out the phosphorescence imaging of supramolecular polymer towards Hela cells with a visible light source (488 nm). The result showed that

the UCNPs + CB[8]/HA–BrBP system could realize the phosphorescence imaging towards cancer cells under the excitation of visible light.

## Discussion

In summary, two pseudorotaxane polymers were constructed by means of host–guest interactions between CBs and HA–BrBP, and these purely organic materials showed RTP in aqueous solution. More importantly, compared with the nanofibrous CB[7]/HA–BrBP assembly, the spherical CB[8]/HA–BrBP pseudorotaxane polymer exhibited an ultralong RTP lifetime (4.33 ms) with a phosphorescence quantum yield of 7.58%. These properties were attributed to the synergistic effect of the strong host–guest interactions of BrBP with CB[8] and hydrogen-bond of the HA chains, which immobilized the phosphors, suppressing nonradiative decay, promoting intersystem crossing, and shielding the triplet state of the phosphor from collisions with triplet oxygen or other molecules. Because the biaxial pseudorotaxane polymer CB[8]/HA–BrBP combined both phosphorescence emission and tumor-cell-targeting ability, it could be used for phosphorescence imaging of mitochondria in tumor cells. We believe that the supramolecular strategy described herein not only will provide a way for the development of purely organic compounds that exhibit RTP in aqueous solution but also will broaden the applications of phosphorescence.

## Methods

**Reagents and materials**. All chemicals were obtained from commercial suppliers, unless noted otherwise. 4-(4-Bromophenyl)pyridine was purchased from Bide Pharmatech, and 3-bromopropan-1-amine was purchased from Struchem Co. NMR spectra were recorded on a Bruker AV400 spectrometer. UCNPs (7.5 mg/ml, NaYREF$_4$, RE: Yb, Er, Tm, Gd, Mu, Lu) was purchased from Hefei Fluonano Biotech Co., Ltd. Fluorescence spectra were recorded in a conventional quartz cell (light path, 10 mm) on a Varian Cary Eclipse spectrophotometer equipped with a Varian Cary single-cell Peltier accessory to control the temperature. UV–vis spectra and optical transmittance were recorded in a quartz cell (light path, 10 mm) on a Shimadzu UV–3600 spectrophotometer equipped with a PTC–348WI temperature controller. Steady-state fluorescence emission spectra were recorded in a conventional quartz cell ($10 \times 10 \times 45$ mm) at 25 °C on a Varian Cary Eclipse spectrophotometer equipped with a Varin Cary single-cell Peltier accessory to control the temperature. Photoluminescence spectra and fluorescence and phosphorescence lifetimes were measured by means of time-correlated single-photon counting on a FLS980 instrument (Edinburg Instruments, Livingstone, UK). High-resolution transmission electron microscopy images were acquired using a Tecnai 20 high-resolution transmission electron microscope operating at an accelerating voltage of 200 keV; the sample was prepared by dropping the solution onto a copper grid, which was then air-dried. Scanning electron microscopy images were obtained with a Hitachi S–3500 N scanning electron microscope. The zeta potentials were determined on a NanoBrook 173Plus at 25 °C. Electrospray ionization mass spectra were measured with an Agilent 6520 Q–TOF–MS instrument. Microsoft 2013 and OriginPro 2020b were used for data analysis.

**ITC measurements**. ITC measurements were performed with an isothermal titration microcalorimeter (VP–ITC, Microcal Inc.) at atmospheric pressure and 25.00 °C in aqueous solution to obtain stability constants ($K_S$) and thermodynamic parameters. A solution of BrBP–NH$_2$ in a 0.250 mL syringe was sequentially injected into a stirring (300 rpm) solution of CB[7] and CB[8] in the sample cell (1.4227 mL volume). The concentrations of CB[7] and BrBP–NH$_2$ were used as 0.048 and 1.25 mM, respectively. The thermodynamic parameters were obtained by using a model with one set of binding sites. The concentrations of CB[8] and BrBP–NH$_2$ were used as 0.040 and 1.28 mM, respectively. The thermodynamic parameters reported in this work were obtained by using a sequential binding sites model with 1:2 stoichiometry.

**Quantum mechanical calculations**. Quantum mechanical calculations were carried out with the Gaussian 16 program[47]. Geometry optimization was performed with the M06-2X-GD3[48] functional and the 6–31 G (d,p) basis set. The single-point energy, Mulliken charge and the energies of the frontier molecular orbitals were calculated at the M06-2X-GD3/6-311 G (d,p) level in water with the SMD solvation model[49]. The optimized structures were rendered using CYLView 1.0b software. Non-covalent interaction (NCI) analysis with an independent gradient model (IGM)[50] method was carried out by Multiwfn 3.7[51] and was rendered using VMD 1.9.3[52].

**General cell culture and imaging**. The cancer cell line including human lung adenocarcinoma cells (A549), human cervical cancer cells (HeLa) as well as human esophageal cancer cells (KYSE-150) and the normal cell line including human renal epithelium cell line (293T) as well as human embryonic lung fibroblast (MRC-5) were all obtained from the Cell Resource Center of China Academy of Medical Science (Beijing, China). The well-cultured cells were incubated with CB[8]/HA–BrBP ([BrBP] = 25 μM, [CB[8]] = 12.5 μM) for 12 h. The cells were then washed with PBS and stained with MitoTracker Red (100 nm, Sigma) at 37 °C for 30 min. Then the cells were further washed three times with phosphate buffer solution, fixed with 4% paraformaldehyde for 15 min, and then observed with a confocal microscope. Confocal images were acquired with 405 nm laser/ 515–540 nm filter and 488 nm laser/ 515–540 nm filter. All the microscope settings were kept consistent in each experiment.

**Cell viability assay**. To evaluate the cytotoxicity of HA–BrBP(G) and CB[8]/HA–BrBP(H + G), the well-cultured cells were treated with different concentrations of HA–BrBP and CB[8]/HA–BrBP for 12 h. The relative viability was determined by standard CCK-8 assay.

**Statistics and reproducibility**. Each experiment was performed with three replicates. Each measurement was taken from three distinct samples. The results indicate means ± standard deviation (SD). Statistical analysis for comparing two experimental groups was performed using two-sided Student's $t$-test ($P < 0.05$). Statistical tests were performed by the SPSS software (version 20, IBM, USA).

**Reporting summary**. Further information on research design is available in the Nature Research Reporting Summary linked to this article.

## Data availability

The authors declare that the data supporting the findings of this study are available within the paper and its Supplementary Information. All data are available from the authors on reasonable request.

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

## Acknowledgements

This work was financially supported by the National Natural Science Foundation of China (grant nos. 21672113, 21772099, 21861132001, and 21971127).

## Author contributions

Y.L., Y.C., and W.-L.Z. conceived and designed the experiments. W.-L.Z. synthesized and performed the chemical characterization. Q.Y., Z.-X.L., and X.-Y.D. conducted biological experiments. H.Z. and J.-J.L. performed density functional theory. W.-L.Z. wrote the main manuscript. Y.L. supervised the work and edited the manuscript. All authors analyzed and discussed the results and reviewed the manuscript.

## Competing interests

The authors declare no competing interests.
