## [Peer Review File · Nature Communications]

Reviewers' Comments:

Reviewer #1:

Remarks to the Author:

This manuscript reports an experimental approach to achieve highly efficient room temperature phosphorescence (RTP) with a long phosphorescence lifetime in aqueous phase. Inspired by author's previous study, they developed a novel supramolecular assembly (i.e. CB[8]/HA-BrBP) which showed a rather bright RTP ($\Phi_p = 7.6\%$) with a long phosphorescence lifetime ($\tau_p = 4.3$ ms) in aqueous phase. Moreover, authors performed combined experimental and computational studies to validate the origin of bright ultra-long RTP of CB[8]/HA-BrBP. Targeted imaging of cancer cells has also been conducted based on the developed material system. The work appears to have been carefully conducted and is also interesting. However, due to previous works in this research field, this reviewer believes that this contribution does not present enough novelty and originality required for publication in Nature Communications.

(1) Currently, many strategies to achieve highly efficient RTP in aqueous phase have been published especially based on cucurbit[n]uril and/or cyclodextrin-mediated supramolecular assembly, which greatly reduces the novelty of this work. Please see the listed papers.

- Scypinski, S. et. al. *Anal. Chem.* 56, 322-327 (1984)
- Zhang, G. et. al. *Nat. Mater.* 8, 747-751 (2009)
- Gong, Y. et. al. *ChemPhysChem* 17, 1934-1938 (2016)
- Xu, L. et. al. *Dyes Pigm.* 142, 300-305 (2017)
- Yu, Y. et. al. *Angew. Chem. Int. Ed.* 56, 16207-16211 (2017)
- Kuila, S. et. al. *Angew. Chem. Int. Ed.* 57, 17115-17119 (2018)
- Yu, X. et. al. *Chem. Commun.* 55, 3156-3159 (2019)
- Wang, J. et. al. *Angew. Chem. Int. Ed.* 59, 1-7 (2020)

(2) This reviewer do not fully agree with author's suggested mechanism for phosphorescence behaviour of CB[8]/HA-BrBP. Since halogen bonding is highly directional (R-X-Y angle tends toward 180; Orangi, N. et. al. *ChemPhysChem*, 20, 1922-1930 (2019)), it is very difficult to make halogen bonding between the Br atom of BrBP-NH₂ and the carbonyl group of CB[8].

(3) Although there exists a halogen bonding between the Br atom of BrBP-NH₂ and the carbonyl group of CB[8] as authors argued, it is very difficult to understand how such a halogen bond facilitates ISC of the phosphor; in fact, the carbonyl group is not a part of emitting molecule (BrBP-NH₂) and thus, ISC cannot be facilitated.

(4) The absorption of the developed system (i.e. CB[8]/HA-BrBP) is in the UV region, making it extremely difficult to utilize for actual cell imaging.

Reviewer #2:

Remarks to the Author:

In this work, Prof. Liu and coworkers reported water-soluble ultralong organic room-temperature phosphorescence supramolecular polymers constructed by cucurbit[n]uril (CB[7], CB[8]) and hyaluronic acid (HA) as a tumor-targeting ligand conjugated to a 4-(4-bromophenyl)pyridin-1-ium bromide (BrBP) phosphor. Owing to the more stable 1:2 inclusion complex between CB[8] and BrBP and to the networks of hydrogen-bond involving HA, this supramolecular polymer had purely organic RTP lifetime in water up to 4.33 ms with a quantum yield of 7.58%. This supramolecular polymer was successful applied for cancer cell targeted phosphorescence imaging of mitochondrion. The results are quite interesting and could be accepted for publication after major revision. 1) The authors should provide more data about different HA polymer distributions on the detection performance. This reviewer suggests the authors should delete some words such as 'nanoparticles' for the supramolecular polymers (if so, the authors should provide the data about

the nanoparticles?). What's aggregates for the cancer detection or which aggregates would have special target? Please delete 'longest' lifetimes of RTP even for water system. 2) The proposed mechanism shown in Fig. 5b did NOT yet demonstrate the CB rigid to enhanced ISC. Please improve this figure. Actually, this reviewer believed the ISC efficiency might be also enhanced by hydrogen-bonding in this system. Please discuss this key point. One most recent related publication should be cited: J. Wang, et al., *Angew. Chem. Int. Ed.* 2020, 10.1002/anie.201914513

Reviewer #3:

Remarks to the Author:

The authors developed organic phosphor nanoparticles having a long phosphorescence emission in aqueous solution. The reported system consists of cucurbituril (CB), BrBP, and hyaluronic acid. By employing NMR, UV absorption, and ITC titration the authors investigated the pseudorotaxane formation between CB and BrBP.

The authors emphasized the value of the achieved persistent phosphorescence emission in aqueous solution. They claim that the strong host-guest interactions suppresses non-radiative relaxation, halogen bonding between CBs and BrBP facilitates ISC, and hydrogen bonding networks among HA suppresses further vibrational relaxation.

The vibrational suppression by hydrogen bonding and host-guest interactions are convincing. The suggested vibrational suppression can be supported by experiments. This reviewer strongly suggest a photoluminescence analysis of HA-BrBP in liquid nitrogen. At this condition vibrational loss should be effectively suppressed and HA-BrBP would produce strong phosphorescence.

However, the attribution to halogen bonding is problematic. Halogen bonding is usefully utilized in crystal design since halogen bonding is directional unlike hydrogen bonding. It appears that the carbonyl group and bromophenyl unit are rather parallel and thereby halogen bonding is not quite feasible considering this spatial arrangement of the bromophenyl unit of BrBP and the carbonyl moieties of CB.

This reviewer read the references 18 and 19 that are the authors' prior publications and closely related to the content of current manuscript. They reported a very high quantum yield from bromophenyl methyl pyridinium chloride/CB[6]. I am wondering why the authors did not use CB[6] for this study rather than CB[7] and CB[8]. The authors contributed the higher QY of CB[8]/HA-BrBP to strong pi-pi stacking. However, wouldn't the pi aggregation cause emission quenching? Even though it is stated that CB[8]/HA-BrBP is brighter than CB[7]/HA-BrBP, the data in Figure 3 (b) show exactly opposite. Is Figure 3(b) a normalized plot?

There are many mistakes found in the manuscript. English revision is also recommended to clear up grammatical glitches. Examples are as follows:

The first "4-phenyl-1-methylpyridin-1-ium chloride" should be "4-(4-bromophenyl)-1-methylpyridin-ium chloride" in line 44 and 45.

The "successful" should be "successfully" in line 27.

The lifetime in line 185 should be 77.08 ms not second.

Reference 22 should be 2019 not 2018.

Reply to Referee 1.

Comments:

This manuscript reports an experimental approach to achieve highly efficient room temperature phosphorescence (RTP) with a long phosphorescence lifetime in aqueous phase. Inspired by author's previous study, they developed a novel supramolecular assembly (i.e. CB[8]/HA-BrBP) which showed a rather bright RTP ($\Phi_p = 7.6\%$) with a long phosphorescence lifetime ($\tau_p = 4.3$ ms) in aqueous phase. Moreover, authors performed combined experimental and computational studies to validate the origin of bright ultra-long RTP of CB[8]/HA-BrBP. Targeted imaging of cancer cells has also been conducted based on the developed material system. The work appears to have been carefully conducted and is also interesting. However, due to previous works in this research field, this reviewer believes that this contribution does not present enough novelty and originality required for publication in Nature Communications.

Reply: We greatly appreciate the referee's comments and useful advice. As the reviewer noted, this work is the system developed by previous work (references 20 and 21). In previous research, the solid-state supramolecular assembly mediated by CB[6] just displayed very weak fluorescence and phosphorescence in aqueous solution due to the quenching of water molecule. On the other hand, although the aqueous phosphorescence had a great significant development (see *Angew. Chem. Int. Ed.* 2020, 59, DOI:10.1002/anie.201914513; *Chem. Sci.*, 2020, 11, 833-838), millisecond-level RTP from purely organic materials in water has rarely been reported. This greatly limits its functionalization and applications. So the realization purely organic room temperature phosphorescence by supramolecular means is helpful to realize the functionalization and regulation of aqueous phosphorescence materials. Therefore, the purely organic phosphorescence with long life (millisecond-level) in water still faces great opportunities and challenges up to now. In this paper, we modified the organic phosphor 4-(4-bromophenyl)-pyridin-1-ium to hyaluronic acid (HA) with cancer cell targeting property, and the effects of cucurbiturils on the phosphorescence of HA-BrBP were systematically studied. Significantly, the resultant CB[8]/HA-BrBP achieved the ultralong RTP lifetime (4.33 ms). Therefore, we believe that the present study should attract much attention of a broad field of scientists who

are interested in supramolecular chemistry, especially the application in the biology system, and therefore merit publication in Nature communications. In addition, we also address the referee's comments as follows:

Q1: Currently, many strategies to achieve highly efficient RTP in aqueous phase have been published especially based on cucurbit[n]uril and/or cyclodextrin-mediated supramolecular assembly, which greatly reduces the novelty of this work. Please see the listed papers.

-Scypinski, S. et. al. *Anal. Chem.* 56, 322-327 (1984)

-Zhang, G. et. al. *Nat. Mater.* 8, 747-751 (2009)

-Gong, Y. et. al. *ChemPhysChem* 17, 1934-1938 (2016)

-Xu, L. et. al. *Dyes Pigm.* 142, 300-305 (2017)

-Yu, Y. et. al. *Angew. Chem. Int. Ed.* 56, 16207-16211 (2017)

-Kuila, S. et. al. *Angew. Chem. Int. Ed.* 57, 17115-17119 (2018)

-Yu, X. et. al. *Chem. Commun.* 55, 3156-3159 (2019)

-Wang, J. et. al. *Angew. Chem. Int. Ed.* 59, 1-7 (2020)

Reply: We thank the referee's good advice. According to the referee's advice, we cited the corresponding references in the revised manuscript. In fact, up to now, although the phosphorescence in water has made great progress through the host-guest interactions, the purely organic phosphorescence with long life (millisecond-level) and function in water still faces great opportunities and challenges. In this paper, we modified the organic phosphor 4-(4-bromophenyl)-pyridin-1-ium to hyaluronic acid (HA) with cancer cell targeting property, and the effects of cucurbiturils on the phosphorescence of HA-BrBP were systematically studied. Significantly, the resultant CB[8]/HA-BrBP achieved the ultralong RTP lifetime (4.33 ms). We believe that these results could provide a novel approach for the application of aqueous phosphorescence supramolecular assembly in biology system.

Q2: This reviewer do not fully agree with author's suggested mechanism for phosphorescence behaviour of CB[8]/HA-BrBP. Since halogen bonding is highly directional (R-X-Y angle tends toward 180; Orangi, N. et. al. *ChemPhysChem*, 20, 1922-1930 (2019)), it is very difficult to make halogen bonding between the Br atom of BrBP-NH₂ and the carbonyl group of CB[8].

Reply: We thank the referee's good advice. According to the reviewer's advice, we rechecked the molecular model of CB[8]/HA-BrBP and found that the halogen bonding existed between the C-Br of BrBP and the N atom of the adjacent BrBP (C-Br...N

angle of 159.6° , Br \cdots N distance 3.09 Å) based on the DFT optimization results (Fig. 5a) and the noncovalent interaction (NCI) analysis (Supplementary Fig. 13). Meanwhile, the Mulliken charge (Supplementary Table 2) on the Br atom also increased after the formation of aggregate. This phenomenon indicated that the C-Br \cdots N halogen bonding changed the charge distribution on Br atom, which would affect the heavy atom effect and thus promote the phosphorescence. In addition, the halogen bonds accompanying with the strong host-guest interactions between CB and BrBP(s) as well as the multiple hydrogen bonds locked the BrBP unit(s) in different directions, probably restricting the molecular motion and reducing the non-radiative decay. These factors jointly contribute to the RTP behavior of CB[8]/HA-BrBP. The corresponding statement was added in the revised manuscript (line 12, Page 15).

Q3: Although there exists a halogen bonding between the Br atom of BrBP-NH₂ and the carbonyl group of CB[8] as authors argued, it is very difficult to understand how such a halogen bond facilitates ISC of the phosphor; in fact, the carbonyl group is not a part of emitting molecule (BrBP-NH₂) and thus, ISC cannot be facilitated.

Reply: We thank the referee's good advice. According to the reviewer's advice, we rechecked the molecular model of CB[8]/HA-BrBP and found that the halogen bonding existed between the C-Br of BrBP and the N atom of the adjacent BrBP (C-Br \cdots N angle of 159.6° , Br \cdots N distance 3.09 Å) based on the DFT optimization results (Fig. 5a) and the noncovalent interaction (NCI) analysis (Supplementary Fig. 13). Meanwhile, the Mulliken charge (Supplementary Table 2) on the Br atom also increased after the formation of aggregate. This phenomenon indicated that the C-Br \cdots N halogen bonding changed the charge distribution on Br atom, which would affect the heavy atom effect and thus promote the phosphorescence. We also calculated the rate constants of radiative decay, nonradiative decay and intersystem crossing (Table 1), and the result demonstrated the enhancement of ISC and the suppressing of nonradiative decay. The corresponding statement was added in the revised manuscript (Table 1, Page 14; line 12, Page 15) and revised Fig 5.

Q4: The absorption of the developed system (i.e. CB[8]/HA-BrBP) is in the UV region, making it extremely difficult to utilize for actual cell imaging.

Reply: We thank the referee's good advice. It is well documented that the cell imaging by the ultraviolet excitation is also an available method (Chem. Mater., 2019, 31, 9887-9894; Chem. Sci., 2020, 11, 419-428; Angew. Chem. Int. Ed. 2020, DOI:10.1002/anie.201914513; Chem. Sci., 2020, 11, 833-838; Small, 2020, 1906733),

despite its penetrability is relatively weak.

Reply to Referee 2.

Comments:

In this work, Prof. Liu and coworkers reported water-soluble ultralong organic room-temperature phosphorescence supramolecular polymers constructed by cucurbit[n]uril (CB[7], CB[8]) and hyaluronic acid (HA) as a tumor-targeting ligand conjugated to a 4-(4-bromophenyl)pyridin-1-ium bromide (BrBP) phosphor. Owing to the more stable 1:2 inclusion complex between CB[8] and BrBP and to the networks of hydrogen-bond involving HA, this supramolecular polymer had purely organic RTP lifetime in water up to 4.33 ms with a quantum yield of 7.58%. This supramolecular polymer was successful applied for cancer cell targeted phosphorescence imaging of mitochondrion. The results are quite interesting and could be accepted for publication after major revision.

Reply: We greatly appreciate the referee's positive comments and useful advice. Herein, we address the referee's comments as follows:

Q1: The authors should provide more data about different HA polymer distributions on the detection performance. This reviewer suggests the authors should delete some words such as 'nanoparticles' for the supramolecular polymers (if so, the authors should provide the data about the nanoparticles?). What's aggregates for the cancer detection or which aggregates would have special target? Please delete 'longest' lifetimes of RTP even for water system.

Reply: We thank the referee's good advice. According to the referee's advice, we investigated the effect of the different HA polymer distributions on the detection performance of aggregates by changing either the molecular weight of the HA polymer skeleton or the ratios between CB[8] and BrBP moiety. The results showed that, for the aggregates obtained from HA with molecular weights of 3kDa, 100kDa and 1000kDa (named CB[8]/HA_{3k}-BrBP, CB[8]/HA_{100k}-BrBP and CB[8]/HA_{1000k}-BrBP respectively), the detection abilities of aggregates towards A549 cells enhanced with the increase of the molecular weight of the HA polymer skeleton, i.e. CB[8]/HA_{3k}-BrBP < CB[8]/HA_{100k}-BrBP < CB[8]/HA_{1000k}-BrBP. However, all of CB[8]/HA_{3k}-BrBP, CB[8]/HA_{100k}-BrBP and CB[8]/HA_{1000k}-BrBP displayed the very weak detection abilities towards normal human cells MRC-5 (human embryonic lung

fibroblast). For the aggregates obtained from the same HA polymer skeleton and the various CB[8] : BrBP ratios (CB : BrBP = 0 equiv. CB[8] : 1 equiv. BrBP, 0.09 equiv. CB[8] : 1 equiv. BrBP and 0.5 equiv. CB[8] : 1 equiv. BrBP), the phosphorescence intensity and cell imaging effect increased with the increase of the CB: BrBP ratio, i.e. free HA-BrBP (CB[8]:BrBP = 0 equiv. CB[8] : 1 equiv. BrBP) < CB[8]/HA-BrBP at 0.09 equiv. CB[8] : 1 equiv. BrBP < CB[8]/HA-BrBP at 0.5 equiv. CB[8] : 1 equiv. BrBP (Supplementary Fig. 17, 18, 19; line 20, Page 19).

In addition, we deleted the word “longest” and changed the word “nanoparticles” to “aggregates”. Moreover, it is documented that the aggregates in the size range of 20–200 nm tend to accumulate in tumour tissues much more than they do in normal tissues (i.e. Chem. Soc. Rev., 2017, 46, 3830-3852; J. Am. Chem. Soc. 2018, 140, 4945-4953), and the size of CB/HA-BrBP aggregates reported by us is within this range. The corresponding statements were added in the revised manuscript.

Q2: The proposed mechanism shown in Fig. 5b did NOT yet demonstrate the CB rigid to enhanced ISC. Please improve this figure. Actually, this reviewer believed the ISC efficiency might be also enhanced by hydrogen-bonding in this system. Please discuss this key point. One most recent related publication should be cited: J. Wang, et al., Angew. Chem. Int. Ed. 2020, 10.1002/anie.201914513

Reply: We thank the referee’s good advice. According to the referee’s advice, we revised Fig.5 in the revised manuscript. In addition, by using a reference method (Chem. Sci., 2019, 10, 7773-7778; Chem. Sci., 2020, 11, 3531-3537), we also calculated the rate constants of radiative decay, nonradiative decay and intersystem crossing (Table 1), and the result demonstrated the enhancement of ISC and the suppressing of nonradiative decay. According to our experimental results and recently reports (Angew. Chem. Int. Ed. 2020, 10.1002/anie.201914513; Chem. Sci., 2020, 11, 3531-3537), we could deduce the possible reason may be: First, in aqueous solution, CBs provides a hydrophobic environment that protects the triplet state from the collision of the triplet oxygen and other molecules. Second, the strong host–guest interactions between CB and BrBP(s) as well as the multiple hydrogen bonds lock the BrBP unit(s) in different directions, probably restricting the molecular motion and reducing the non-radiative decay. Third, comparing with CB[7], CB[8] can accommodate two BrBP moieties, which results in more stable aggregation of CB[8]–enhanced π – π complexes (with a higher association constant) to limit molecular rotation and enhance ISC more efficiently, resulting in a longer phosphorescence lifetime. Fourth, the formation of halogen bonding between the C-Br of BrBP and the N atom of the adjacent BrBP (C-Br \cdots N angle of 159.6°, Br \cdots N distance 3.09 Å) increase the Mulliken charge on the Br atom and thus affected the heavy effect. As a result, the combination of the host–guest interaction, π – π /Br– π interaction, halogen

bonding and multiple hydrogen bonding jointly contributes to the long RTP of CB[8]/HA–BrBP in aqueous solution probably by restricting the molecular motion, promoting the ISC and reducing the non-radiative decay. The corresponding statement was added in the revised manuscript (Table 1, Page 14; line 12, Page 15) and revised Fig 5. In addition, we also cited the related reference (J. Wang, et al., *Angew. Chem. Int. Ed.* 2020, 10.1002/anie.201914513) in the revised manuscript.

Reply to Referee 3.

Comments:

The authors developed organic phosphor nanoparticles having a long phosphorescence emission in aqueous solution. The reported system consists of cucurbituril (CB), BrBP, and hyaluronic acid. By employing NMR, UV absorption, and ITC titration the authors investigated the pseudorotaxane formation between CB and BrBP. The authors emphasized the value of the achieved persistent phosphorescence emission in aqueous solution. They claim that the strong host-guest interactions suppresses non-radiative relaxation, halogen bonding between CBs and BrBP facilitates ISC, and hydrogen bonding networks among HA suppresses further vibrational relaxation. The vibrational suppression by hydrogen bonding and host-guest interactions are convincing. The suggested vibrational suppression can be supported by experiments.

We greatly appreciate the referee's positive comments and useful advice. Herein, we address the referee's comments as follows:

Q1: This reviewer strongly suggest a photoluminescence analysis of HA-BrBP in liquid nitrogen. At this condition vibrational loss should be effectively suppressed and HA-BrBP would produce strong phosphorescence.

Reply: We thank the referee's good advice. According to the referee's advice, we performed a variable temperature experiment of HA-BrBP from 298 K to 100 K, which could effectively suppress vibrational loss in liquid nitrogen. The result showed that HA-BrBP produced the strong phosphorescence as the reviewer noted. We added the corresponding the phosphorescence spectra and lifetime curves of HA-BrBP in Supplementary Fig. 11, and the corresponding statement in the revised manuscript (line 5, Page 14).

Q2: However, the attribution to halogen bonding is problematic. Halogen bonding is usefully utilized in crystal design since halogen bonding is directional unlike hydrogen bonding. It appears that the carbonyl group and bromophenyl unit are rather parallel and thereby halogen bonding is not quite feasible considering this spatial arrangement of the bromophenyl unit of BrBP and the carbonyl moieties of CB.

Reply: We thank the referee's good advice. According to the reviewer's advice, we rechecked the molecular model of CB[8]/HA-BrBP and found that the halogen bonding existed between the C-Br of BrBP and the N atom of the adjacent BrBP (C-Br \cdots N angle of 159.6°, Br \cdots N distance 3.09 Å) based on the DFT optimization results (Fig. 5a) and the noncovalent interaction (NCI) analysis (Supplementary Fig. 13). Meanwhile, the Mulliken charge (Supplementary Table 2) on the Br atom also increased after the formation of aggregate. This phenomenon indicated that the C-Br \cdots N halogen bonding changed the charge distribution on Br atom, which would affect the heavy atom effect and thus promote the phosphorescence. In addition, the halogen bonds accompanying with the strong host-guest interactions between CB and BrBP(s) as well as the multiple hydrogen bonds locked the BrBP unit(s) in different directions, probably restricting the molecular motion and reducing the non-radiative decay. These factors jointly contribute to the RTP behavior of CB[8]/HA-BrBP. The corresponding statement was added in the revised manuscript (line 12, Page 15).

Q3: This reviewer read the references 18 and 19 that are the authors' prior publications and closely related to the content of current manuscript. They reported a very high quantum yield from bromophenyl methyl pyridinium chloride/CB[6]. I am wondering why the authors did not use CB[6] for this study rather than CB[7] and CB[8].

Reply: We thank the referee's good advice. According to the referee's advice, we added the phosphorescence experiment using CB[6]/HA-BrBP (Supplementary Fig. 8). The results showed that the phosphorescence of CB[6]/HA-BrBP was a little stronger than that of CB[7]/HA-BrBP but much weaker than that of CB[8]/HA-BrBP under the same condition. Considering the fairly lower water solubility of CB[6] than those of CB[7] and CB[8] under our experimental condition, we chose CB[7] and CB[8] to conduct systematic optical research. The corresponding statement was added in the revised manuscript (line 10, Page 12).

Q4: The authors contributed the higher QY of CB[8]/HA-BrBP to strong pi-pi stacking. However, wouldn't the pi aggregation cause emission quenching? Even though it is stated that CB[8]/HA-BrBP is brighter than CB[7]/HA-BrBP, the data in

Figure 3 (b) show exactly opposite. Is Figure 3(b) a normalized plot?

Reply: We thank the referee's good advice. According to the referee's advice, we calculated the rate constants of radiative decay (K_r^{Phos}), nonradiative decay (K_{nr}^{Phos}) and intersystem crossing (K_{isc}) based on the reported methods (Table 1), and discussed their changes in the revised manuscript. Generally, the relatively small ($K_r^{Phos} + K_{nr}^{Phos}$) value is beneficial for achieving a long lifetime, while the high Φ_{Phos} value is the comprehensive result of efficient ISC (high Φ_{isc}) and high efficiency of phosphorescence (high $K_r^{Phos} / (K_r^{Phos} + K_{nr}^{Phos})$). In our work, the ($K_r^{Phos} + K_{nr}^{Phos}$) values of CB[8]/HA-BrBP (or CB[8]/BrBP-NH₂) was smaller than that of CB[7]/HA-BrBP (or CB[7]/BrBP-NH₂), and the ($K_r^{Phos} + K_{nr}^{Phos}$) value of CB[8]/HA-BrBP (or CB[7]/HA-BrBP) was smaller than that of CB[8]/BrBP-NH₂ (or CB[7]/BrBP-NH₂), respectively. Meanwhile, the $K_r^{Phos} / (K_r^{Phos} + K_{nr}^{Phos})$ value of CB[8]/HA-BrBP was the largest one among the corresponding values of four assemblies. This indicated the enhancement of ISC and the inhibition of nonradiative decay from the halogen bond, host-guest interaction and hydrogen-bonding interaction enhanced the lifetime and QY. The corresponding statement was added in the revised manuscript (Table 1, Page 14; line 12, Page 15). Our experimental results also demonstrated the strong π - π stacking within molecular dimers could alter the lifetimes of triplet excitons through decreasing their radiative and nonradiative rates (Ref. Nat. Mater., 2015, 14, 685-690; Nat. Photonics, 2019, 13, 373-375; Nat. Commun., 2018, 9, 840; Mater. Chem. Front., 2019, 3, 1391-1397; Chem. Mater., 2014, 26, 2467-2477). When the ISC was enhanced and the nonradiative decay was suppressed, it's going to decrease the electrons in S1 and increase the electrons in T1, and thus to decrease the fluorescence. That is, the π - π stacking will cause the fluorescence emission quenching but improve the phosphorescence emission for the suppressing nonradiative decay, like the recently reported contribution of π - π stacking in the phosphorescence system (Chem. Sci., 2020, 11, 833-838).

In addition, Figure 3(b) isn't a normalized plot, but we are sorry for that Figure 3(b) gave a confused color tag. Indeed, CB[8]/HA-BrBP is exactly brighter than CB[7]/HA-BrBP in Figure 3(b). Accordingly, we modify the Figure 3(b) in the revised manuscript to make it more clearly.

Q5: There are many mistakes found in the manuscript. English revision is also recommended to clear up grammatic glitches. Examples are as follows: The first "4-phenyl-1-methylpyridin-1-ium chloride" should be "4-(4-bromophenyl)-1-methylpyridin-ium chloride" in line 44 and 45. The "successful" should be "successfully" in line 27. The lifetime in line 185 should be 77.08 ms not second. Reference 22 should be 2019 not 2018.

Reply: We thank the referee's good advice. According to the referee's advice, we corrected the mistakes in the manuscript.

Reviewers' Comments:

Reviewer #1:

Remarks to the Author:

In the revised manuscript, the authors mostly addressed the technical issues raised by the reviewers. But this reviewer is still suspicious of the effect of halogen bonding of "C-Br...N", more specifically, the effect of increasing Mulliken charge of bromine to the increase of ISC.

In any case, this reviewer is still reluctant about the acceptance of this manuscript in this journal. This manuscript did not deliver a new/novel design concept that will give an insightful message to this research field nor extremely high phosphorescence QY. Moreover, the authors still use UV-light as an excitation light which is revealed to be harmful to biosystem and thereby, cannot be utilized to real applications.

Therefore, this reviewer cannot recommend the publication of this manuscript to this journal.

Reviewer #2:

Remarks to the Author:

This revision could be accepted.

Reply to Reviewer 1.

Comments:

In the revised manuscript, the authors mostly addressed the technical issues raised by the reviewers. But this reviewer is still suspicious of the effect of halogen bonding of "C-Br...N", more specifically, the effect of increasing Mulliken charge of bromine to the increase of ISC.

Reply: We thank the reviewer's comments. Indeed, the DFT calculation on the molecular model of CB[8]/HA-BrBP demonstrates that the halogen bonding exists between the C-Br of BrBP and the NH₂ of the adjacent BrBP in the angle of 159.6° (Fig. 5a), and the Mulliken charge (Supplementary Table 2) on the bromine group increased slightly after the formation of halogen bond. Moreover, the structural and the noncovalent interaction (NCI) analysis (Supplementary Fig. 13) show that, in addition to the halogen bond, there also exist the host-guest interaction, π - π /Br- π interaction and multiple hydrogen bond in the supramolecular polymer. The combination of the host-guest interaction, π - π /Br- π interaction, halogen bonding and multiple hydrogen bonding jointly contribute to the long RTP of CB[8]/HA-BrBP in aqueous solution probably by restricting the molecular motion, promoting the ISC and reducing the non-radiative decay, which is verified by the calculated radiative and nonradiative decay rate constants. The corresponding statement is highlighted in the manuscript.

In any case, this reviewer is still reluctant about the acceptance of this manuscript in this journal. This manuscript did not deliver a new/novel design concept that will give an insightful message to this research field nor extremely high phosphorescence QY.

Reply: We thank the reviewer's comments, and our manuscript provides a new synergistic enhancement strategy to realize ultralong purely organic aqueous phosphorescence supramolecular polymer for targeted tumor cell. In fact, our manuscript reports a supramolecular polymer with good water solubility, purely organic room temperature phosphorescence with millisecond-level long life, and good cancer cell targeting property, which is very rare in the previous reports.

Moreover, the authors still use UV-light as an excitation light which is revealed to be harmful to biosystem and thereby, cannot be utilized to real applications.

Reply: We thank the reviewer's comments. Although the reviewer 1 states that the use of UV-light as an excitation light is harmful to biosystem and cannot be utilized to real applications, this problem can be solved by the up-conversion luminescence or chemiluminescence method. Accordingly, we try to address this problem by the up-conversion luminescence method (Supplementary Figure 20). As shown in Supplementary Figure 20b, after the addition of the up-conversion nanoparticles (UCNPs) to the supramolecular polymer, the photoluminescence spectrum of supramolecular polymer in water shows a clear emission peak at 510 nm assigned to the phosphorescence emission of CB[8]/HA-BrBP when excited by near infrared light (980 nm). In the control experiment, the UCNPs show no any appreciable emission over 500 nm under the same condition. Moreover, because the confocal microscopy imaging with a near infrared light source is difficult to be performed during the Covid-19 pandemic outbreak, we carry out the phosphorescence imaging of supramolecular polymer towards cancer cells with a visible light source (488 nm). The result shows that the UCNPs+CB[8]/HA-BrBP system can realize the phosphorescence imaging towards cancer cells under the excitation of visible light. These results can thus solve the problem raised by the reviewer 1.

Supplementary Figure 20. The Prompt photoluminescence contrast spectra of UCNPs and UCNPs/CB[8]/HA-BrBP. (a) The excitation spectra of the CB[8]/HA-BrBP upon photoirradiation in aqueous solution at 298 K. (b) The prompt photoluminescence contrast spectra of UCNPs (black) and UCNPs/CB[8]/HA-BrBP (red) in water (298 K, $\lambda_{\text{ex}} = 980$ nm). (c) Confocal microscopy images of HeLa cells incubated with UCNPs/CB[8]/HA-BrBP ($\lambda_{\text{ex}} = 488$ nm).

Therefore, this reviewer cannot recommend the publication of this manuscript to this journal.

Reply: According to the above statements, we added some new experiments and replied the reviewers' all comments one by one, and our present manuscript could be acceptable for publication in Nature Communications.

Reviewer #2 (Remarks to the Author):

This revision could be accepted.

Reply: We greatly appreciate the reviewer's very positive comments.

Reviewers' Comments:

Reviewer #1:

Remarks to the Author:

All concerns previously raised by the reviewers are now nicely addressed. And thus, this reviewer suggest the acceptance of this manuscript in Nat. Commun.